# The Effects of New Urbanization Pilot City Policies on Urban Innovation: Evidence from China

Shengsheng Li [1], Yuanyuan Wang [2,*], Hasan Dincer [3,*], Serhat Yuksel [3] and Dongyao Yu [4]

1   School of Business, Fuyang Normal University, Fuyang 236037, China; lisheng2@foxmail.com
2   School of Economics and Management, Tianjin University of Science and Technology, Tianjin 300222, China
3   School of Business, Istanbul Medipol University, Istanbul 34810, Turkey; serhatyuksel@medipol.edu.tr
4   School of Economics, Beijing Technology and Business University, Beijing 100048, China; ydy0727@126.com
*   Correspondence: 2019110112@email.cufe.edu.cn (Y.W.); hdincer@medipol.edu.tr (H.D.)

**Abstract:** The new urbanization city pilot policy is China's most recent policy on urban urbanization. This paper uses new urbanization pilot policies as a quasi-natural experiment to empirically test the impact of new urbanization pilot policies on urban innovation through the difference-in-differences (DID) method using panel data from 199 cities in China from 2011 to 2019. The results show that: (1) The new urbanization city pilot policy has significantly enhanced urban innovation. (2) The theoretical mechanism test shows that the pilot policy of new urbanization promotes urban innovation through the level of human capital. (3) The results of the heterogeneity analysis show that the new urbanization pilot policies have obvious city-level heterogeneity and regional heterogeneity on the improvement of urban innovation levels. The impact effect of new urbanization pilot policies is higher in first-tier and second-tier cities than in fourth-tier and fifth-tier cities; the effect of new urbanization pilot policies is higher in western regions than in eastern and middle regions.

**Keywords:** new urbanization pilot policy; urban innovation; DID; human capital

## 1. Introduction

### 1.1. Background

After the Chinese economic reform in 1978, China ushered in a wave of industrialization centered on economic construction, which led to the rapid development of industry and commerce and greatly accelerated the process of urbanization. At the same time, China's urbanization shifted from encouraging the development of small and medium-sized cities and small towns to full-scale urban expansion. Existing studies have conducted a large amount of detailed research on the content and characteristics of traditional urbanization, and it has been found that the period from the Chinese economic reform to 2012 is called the traditional urbanization stage. The prominent feature of traditional urbanization is the rapid increase in urbanization rate, which reached 52.57% during the traditional urbanization stage, and nearly tripled by 34.67%. China's urbanization achieved rapid development and made significant contributions to the historical process of promoting urbanization in the world. Along with the rapid growth of the urbanization rate, it has also produced a series of negative impacts on the economy and society. For example, the one-sided focus on urban expansion, blindly engaging in "city building" and building new districts, has led to low-quality urbanization, an urbanization level lower than the level of economic and industrial development, the urbanization of land faster than the urbanization of the population, and structural problems such as increased pressure on the transportation network and high local traffic congestion.

If the traditional urbanization development model continues to be adopted, it will further aggravate and trigger economic and social hidden dangers such as environmental degradation, weak consumer demand, and sloppy and inefficient types of investment in China. Therefore, there is an urgent need to draw on advanced international experience,

accelerate the high-quality development of urbanization, take a new type of people-oriented urbanization path, and fully optimize and upgrade traditional urbanization. China's new urbanization pays more attention to the level of urban public services, ecological and environmental quality, and people-oriented new urbanization [1]. To this end, since 2015, China has set up three batches of new-type urbanization comprehensive reform pilots to explore the transformation path of urbanization from focusing on quantitative scale increase to quality connotation improvement. The Chinese government has successively introduced relevant policies to support and accelerate the construction of new urbanization over the past six years. As of 2020, a total of 188 cities (districts, counties, and towns) have been promoted to implement new urbanization in three batches of pilot cities, and the policy of comprehensive pilot cities for new urbanization is beginning to bear fruit, highlighted by the fact that new urbanization has abandoned the past pursuit of simple urban population ratio increase and scale expansion, and has achieved innovation in various aspects such as concepts, institutions, and culture.

New urbanization is the key to unlocking China's domestic demand potential and development momentum. Economist Joseph Eugene Stiglitz once predicted that China's urbanization would be one of the important issues profoundly affecting human development in the 21st century [2]. In 2019, China's urbanization rate exceeded 60% for the first time [3], surpassing the global average and becoming an important engine for promoting high-quality economic development in the new era. The global urbanization rate in 2019 was about 55.5%, and among the six continents with resident populations, North America had the highest level of urbanization at about 82%; in second place was South America at about 81%; Europe was third at about 74%; Oceania was fourth at 68%; Asia was fifth at just under 50%; and Africa was the lowest at 43%. China now has an urbanization of 60.6%, which exceeds the global average and is higher than the Asian and African averages, but is currently lower than North and South America, Europe, and Oceania.

China's traditional urbanization development path takes a decentralized, low-density, and sloppy form [4,5], which is highlighted by the implementation of a material-oriented development concept, equating urbanization construction with town construction, focusing unilaterally on urban-scale expansion, blindly engaging in "city-building" and building new districts [6], and neglecting the simultaneous development of industrial agglomeration and population agglomeration, leading to problems such as the lack of close integration of production and urbanization, the relatively late integration of migrant workers, and the insufficient supply capacity of public services. If China continues to adopt the traditional urbanization development model, it will further aggravate and trigger economic and social pitfalls such as environmental degradation, weak consumer demand, and sloppy and inefficient investment types in China. Therefore, there is an urgent need to take a new road of people-oriented urbanization and to fully optimize and upgrade traditional urbanization.

Compared with traditional urbanization, the new type of urbanization has a richer connotation, mainly in the following ways: First, the new type of urbanization guarantees and respects the various rights of urban residents [7], accelerates the citizenship of the rural floating population, and provides guarantees for the real integration of the transferred population into the towns by encouraging entrepreneurship. Second, the new urbanization requires the implementation of innovation-driven development strategies [8]. On the one hand, this could be achieved by creating an innovative environment, strengthening the construction of innovation infrastructure, building an "information highway" network, and taking the innovation demonstration landscape as a starting point, promoting industrial technology on the fast track to informationization and intelligence. On the other hand, with the help of big data, the Internet, and other information technology, a smart city could be built as the focus of promoting new urbanization [9]. It can be seen that the new urbanization is based on absorbing the experience of traditional urbanization development, using information technology construction, emphasizing urban innovation and sustainable

development, and relying on high-tech industrial development to fully optimize and enhance traditional urbanization.

Urban innovation capability is a manifestation of innovative cities [10], which refers to cities with strong independent innovation capabilities, outstanding roles in supporting and leading science and technology, high levels of sustainable economic and social development, and significant regional radiation and driving effects. As a regional innovation highland, innovative cities play a crucial role and are important pillars for the construction of innovative countries [11]. At present, innovative cities in China are mainly oriented to prefecture-level cities and are often judged by an evaluation index system consisting of 5 primary indicators and 30 secondary indicators, such as innovation governance, original innovation, technology innovation, achievement transformation, and innovation driving force. In this paper, the total number of patent applications is selected as the measurement index, which can reflect the innovation power of the city.

### 1.2. Research Purpose

In the literature, on the one hand, the evolution of urbanization cannot be separated from industrialization; for example, the first industrial revolution and the second industrial revolution drove the urbanization process in Britain and the United States, respectively [12]. On the other hand, urbanization also drives economic development and becomes an important spatial vehicle for regional economic activities, such as the London metropolitan area, the Tokyo metropolitan area, the Bosworth urban agglomeration, and the Beijing metropolitan area in China. It has been proved that urbanization is a dynamic evolutionary process embedded in the national economic system, and the pattern and level of urbanization are determined by the characteristics of economic development stages and institutional policy systems [13]. The urbanization in Britain was based on the development of the industrial revolution and the lessons learned from the enclosure movement before the industrial revolution, and regulations were enacted to steadily promote urbanization. On the contrary, some Latin American countries have experienced "over-urbanization", in which urbanization has greatly exceeded the level of industrialization and agricultural modernization. In Mexico, Chile, Argentina, Uruguay, and other countries, the urbanization rate has reached 80% or even 90%, but this is a phenomenon of "false urbanization".

Along with the follow-up to the pilot policy practice of new urbanization, the construction of new urbanization has become an important measure to stimulate the innovative vitality of cities. The construction of new urbanization will have a multifaceted promotional effect on improving urban innovation. First, the pilot cities of new urbanization attach importance to building new infrastructure construction, including the creation of big data centers, artificial intelligence, industrial Internet, and many other new digital industry fields, to provide infrastructure for society, enterprises, and financial institutions to improve their innovation power. Second, the pilot cities of new urbanization emphasize the concept of people-oriented development based on the promotion of population urbanization, accelerate the citizenship of rural migrant populations [14], and focus on the development of education and knowledge popularization, which is conducive to improving the level of human capital. The human capital reserve is the think tank of urban development and the powerhouse of innovation [15,16]. Thirdly, in the construction of new urbanization pilot cities, more attention will be paid to industrial upgrading and smart city construction, which will have a catalytic effect on improving the level of urban innovation by accelerating the technological transformation of traditional industries in the city, developing high-tech industries and smart finance, and other developments.

In fact, the construction of China's new urbanization has greatly stimulated the spread of technology levels and knowledge and techniques in regional cities after years of larger-scale pilot and extension construction starting in 2014. Since 2015, China has set up three batches of new urbanization comprehensive reform pilots to explore the transformation path of urbanization from favoring quantitative scale increase to focusing on quality connotation improvement. As of 2020, a total of 188 cities (districts, counties, and towns)

have been promoted in three batches of the pilot city list to implement new urbanization, and the pilot policy of new urbanization is beginning to bear fruit, as highlighted by the fact that new urbanization has rejected the previous pursuit of simple urban population ratio increase and scale expansion. China's new urbanization has achieved innovation and creativity in many aspects, including concepts, institutions, and culture. As the scope of new urbanization pilot cities continues to expand, it is particularly important to scientifically assess the impact of new urbanization construction on urban innovation development.

This paper focuses on assessing the impact of new urbanization pilot policies on urban innovation and its mechanism of action, aiming to address the following questions: First, can new urbanization pilot policies effectively promote the improvement of urban innovation? Second, what is the mechanism of action of the pilot policies of new urbanization in influencing urban innovation? Third, is there regional heterogeneity in the impact of pilot new urbanization construction on urban innovation? The answers to these questions will help to better summarize the experience of new urbanization pilot policy construction and better promote the high-quality development of Chinese cities.

Therefore, this paper proposes a direction for urbanization development in less developed countries or regions, which is to follow the natural laws of urban evolution, improve public infrastructure and public services in cities, and avoid "over-urbanization" and "lagging urbanization". The promotion of new urbanization should be matched with the current economic development, taking the level of economic development as the basis for urbanization, focusing on assessing whether public services are coordinated with the needs of urban populations, and avoiding the speed of upgrading infrastructure and public services lagging behind the speed of urbanization.

*1.3. Research Significance*

In the Chinese literature, there is an initial academic consensus that new urbanization affects innovation, but the extent of the impact of new urbanization innovation and its mechanism of action are not clear. The mechanism through which new urbanization affects the increase of innovation remains a hot topic of debate. The question of whether a series of policy measures adopted during the promotion of China's new urbanization city pilot policy have effectively increased urban innovation is subject to further study. In summary, the following shortcomings remain in existing studies: First, most of the existing domestic studies use regional data at national and provincial levels to measure new urbanization and influence factors, which can hardly reflect the whole picture of China's new urbanization and neglect the impact of new urbanization on micro-cities, resulting in the lack of detailed analysis of the influencing subjects and influence mechanisms. Second, much of the literature overlooks the policy–practice effects produced by the pilot cities of new urbanization, which serve as realistic practice channels to enhance urban innovation and provide quasi-natural experiments to further validate the realistic impacts of new urbanization, thus providing corresponding policy recommendations to continue deepening the construction of new urbanization, optimizing resource allocation, and improving economic efficiency.

How new urbanization affects innovation has become a preliminary consensus in academic circles, but the degree of impact and mechanism of new urbanization innovation are not clear. The mechanism through which new urbanization affects the increase of innovation remains a hot topic of debate. The question of whether a series of policy measures adopted during the promotion of China's new urbanization city pilot policy have effectively increased urban innovation is subject to further study. The following shortcomings remain in the existing studies:

First, most of the existing domestic studies use national and provincial-level regional data to measure new urbanization and impact factors, which can hardly reflect the full picture of new urbanization in China and ignore the impact of new urbanization on micro-cities, resulting in a lack of detailed analysis of impacting subjects and impact mechanisms.

Second, much of the literature ignores the policy practice effects brought on by pilot cities of new urbanization, which serve as a realistic practice channel to enhance urban innovation and provide quasi-natural experiments to further verify the realistic impacts of new urbanization, thus providing corresponding policy suggestions to continue deepening the construction of new urbanization, optimizing resource allocation, and improving economic efficiency.

The value of this paper lies in the following three points: (1) This study can enrich our understanding of new urbanization, accurately grasp the synergistic requirements of China's new urbanization and urban innovation, and provide important policy references for the enhancement of China's urban innovation capacity. (2) On the basis of theoretical and empirical studies, it provides policy recommendations for further promoting the construction of new urbanization and focusing on its economic effects. In the context of the new era, how to enhance the innovation capacity of cities and promote high-quality economic development has become an important practical issue. By promoting the construction of new urbanization, stimulating the inherent potential of new urbanization, and improving infrastructure construction, etc., the Chinese government will be important in improving the innovation capacity of cities. (3) This paper can formulate the key development directions of new urbanization policies for the improvement of the urban innovation capacity in different regions, and further provide realistic and feasible guidance for the improvement of overall innovation capacity.

## 2. Methodology

To analyze the impact of new urbanization construction pilot policies on urban innovation, we refer to Beck et al., 2010 and define the following difference-in-differences (DID) model [17]:

$$y_{i,t} = \alpha + \theta(treat_i \times post_{i,t}) + \beta X_{i,t} + \mu_i + \lambda_t + \varepsilon_{i,t} \quad (1)$$

where $y_{i,t}$ is the dependent variable; $i$ ($i = 1, 2 \cdots, N$) is the individual; $t$ ($t = 1, 2 \cdots, T$) is the time; $\mu_i$ is the individual fixed effect; $\lambda_t$ is the time fixed effect; $X$ is the control variable which changes with time and individual; $\beta$ is the coefficient of the control variable; and $\varepsilon_{i,t}$ is the model error term. The variable *treat* is the dummy variable of the treated group. If individual $i$ belongs to the "treated group" which is impacted by the policy, that is treat = 1; if the individual $i$ belongs to the "control group" which is not impacted by the policy, that is treat = 0 [16,18]. The variable *post* is the dummy variable in the processing period, and the individual in the processing group will not be impacted by the policy until the processing period. If individual $i$ enters the processing period, that is post = 1; otherwise, post = 0.

The cross-multiplication term $treat_i \times post_{i,t}$ in the above model is equivalent to the dummy variable ($did_{i,t}$), which represents individual $i$ being processed in the $t$ period. Therefore, the DID model can also be set as follows [18]:

$$y_{i,t} = \alpha + \theta did_{i,t} + \beta X_{i,t} + \mu_i + \lambda_t + \varepsilon_{i,t} \quad (2)$$

Based on the above analysis, we constructed a DID model to evaluate the policy effects of the pilot policy of new urbanization construction on the level of urban innovation. Since new urbanization construction pilot policies are implemented in different regions and at different times, we built a time-varying DID model for the analysis. The model is established as follows:

$$innovation_{i,t} = \beta_0 + \beta_1 did_{i,t} + \beta_2 X_{i,t} + \mu_i + \lambda_t + \varepsilon_{i,t} \quad (3)$$

where $i$ is the city individual, $t$ is the time, *innovation* is the urban innovation, and $X$ is the control variable. The variable *did* is the constructed difference-in-differences item, that is, the city is included in the pilot policy, and subsequent years are defined as $did = 1$, otherwise $did = 0$. The estimated coefficient $\beta_1$ is the impact effect of new urbanization construction pilot policies on urban innovation.

Since the new urbanization construction pilot policy is officially piloted in different months, where the first batch of pilot cities was officially announced in February 2015, in this paper, we chose to assign 2015 as the policy point in time, assigning the batch of pilot cities 2015 and later as 1, and 0 before 2015; the second batch of pilot cities was announced in November 2015, and we chose 2016 as the policy point in time for assignment, assigning the batch of pilot cities 2016 and later to 1, and 0 before 2016; the third batch of pilot cities was officially announced in December 2016, and we chose to assign 2017 as the policy point in time, assigning the batch of pilot cities 2017 and later to 1, and 0 before 2017.

## 3. Data

(1)    Explained variable

The explained variable in this paper is urban innovation (*innotion*). In this paper, we use the number of city patent applications to represent urban innovation [19]. The number of city patents is processed by adding 1 and taking the natural logarithm (*lntpatent*). The main reason for using the number of invention patent applications as a measure of urban innovation is that patent data are open, objective, and rarely manipulated [16].

(2)    Explanatory variable

The variable *did* is the main explanatory variable, and the estimated coefficient $\beta_1$ represents the policy effect of policies on urban innovation.

(3)    Control variables

Based on the approach of the available literature, the control variables affecting urban innovation were selected as follows.

Level of urban economic development (*lngdp*): measured using the natural logarithm of the city's GDP [18]. Since the level of economic development represents the economic strength of a region, if the level of economic development is higher, it is more likely to increase regional infrastructure investment and R&D investment, which will have a positive promotion effect on the level of urban innovation.

Trade level (*lntrade*): measured using the natural logarithm of the total domestic and foreign trade of each city. The development of international trade has greatly contributed to scientific and technological innovation [20], which has been one of the most important factors for China to gain a foothold in international cooperation since its accession to the World Trade Organization.

Education level (*edu*): measured using years of schooling per capita in each city. Education can provide talents and intellectual resources for scientific and technological innovation. China's higher education and research institutions have produced a large number of scientific and technological talents who have become an important force for scientific and technological innovation. In addition, education can provide a wealth of disciplinary knowledge and skills, which in turn can provide the host city with innovation capital and promote urban innovation.

Infrastructure development (*lnroad*): measured using the natural log of road miles owned by each city. The public services carried by infrastructure are at the heart of cities. Taking the transportation construction in Beijing, Shanghai, and the Guangdong-Hong Kong-Macao Greater Bay Area as examples, infrastructure innovation has become an important force in China's national innovation system [21].

Environmental quality (*lngreen*): measured using the natural logarithm of green space per city. The Chinese government has proposed a policy to protect the environment, and in the process of protecting the environment, environmental technology has integrated green technologies into various industries and fields, and the application of information technology, cloud computing, and big data in the field of environmental protection continues to expand and deepen, promoting technological innovation in cities. In addition, the continuous integration of environmental technology and new technologies further drives the great development of the environmental protection industry.

Financial level (*lndeposit*): measured using the natural logarithm of resident savings in each city. The development of technology enterprises cannot be separated from the support of financial capital, which provides capital support for urban innovation.

The data in this paper were obtained from the China Research Data Service Platform (CNRDS), China City Statistical Yearbook 2012–2020, and the sample period of 2011–2019 was selected in order to retain as much information as possible from the city statistical yearbook. Cities with serious omissions of variables were removed, and the missing values of some variables were filled in by consulting the statistical yearbooks of the provinces or completed using the method of the average annual growth rate of cities. Considering that the study sample was prefecture-level cities, the sample of prefecture-level cities where the pilot counties or cities were located was further excluded in order to avoid bias in the sample grouping due to the prefecture-level city sample where the counties or cities entered the list. Finally, non-parallel panel data of 199 cities with a total of 1789 observations from 2011 to 2019 were obtained, including 47 prefecture-level pilot cities, and were grouped into the treatment group, while the remaining 152 cities were grouped into the control group. The statistics of the variables are shown in Table 1.

**Table 1.** Statistical description of variables.

| Variable | Obs. | Mean | SD | Med | Min | Max |
|---|---|---|---|---|---|---|
| *lntpatent* | 1789 | 4.796 | 1.752 | 4.575 | 0.000 | 9.492 |
| *did* | 1789 | 0.111 | 0.314 | 0.000 | 0.000 | 1.000 |
| *lnpgdp* | 1789 | 10.704 | 0.580 | 10.650 | 8.773 | 12.579 |
| *lntrade* | 1789 | 13.761 | 2.092 | 13.787 | 3.219 | 19.254 |
| *lnedu* | 1789 | 7.211 | 0.459 | 7.159 | 5.796 | 9.002 |
| *lnroad* | 1789 | 0.047 | 0.612 | −0.118 | −0.966 | 2.689 |
| *lngreen* | 1789 | −2.446 | 1.017 | −2.405 | −6.051 | 2.394 |
| *lndeposit* | 1789 | 16.759 | 0.985 | 16.642 | 14.415 | 20.156 |

## 4. Results and Discussion

### 4.1. DID Results Analysis

We use a two-way fixed-effects model to analyze the impact of the new urbanization pilot policy on urban innovation, and the specific regression results are shown in Table 2. In Table 2, columns (1) and (2) do not control for time and individual effects, column (3) controls for time and individual effects, while control variables are added in columns (2) and (3). The results show that in column (3), the estimated coefficient of the difference-in-differences term *did* is significantly positive at the 10% level after simultaneously controlling for time effects, individual effects, and the inclusion of control variables, indicating that the pilot policy of new urbanization construction has significantly increased the level of urban innovation. The estimated coefficient of the difference-in-differences term is 0.0631, indicating that the pilot policy increased the level of urban innovation by 6.31%.

Why have new urbanization construction pilot policies increased the level of urban innovation? The promotion effect of new urbanization on urban innovation capacity mainly comes from three aspects: First, the construction of industrial scientific and technological innovation parks, etc., in the new urbanization construction positively influences the production and operation behavior and infrastructure support of micro enterprises, providing the basic conditions for the innovation of the enterprises in the region. Second, the focus of new urbanization construction also includes developing economies of scale and reducing transaction costs as well as trade costs, such as new urbanization promoting the agglomeration and diffusion of factors through the role of population mobility and resource allocation [22], and improving urban innovation capacity through improving labor quality. Third, new urbanization guides the gradual development of industries in the region from traditional labor-intensive to modern capital-led [23], i.e., it guides enterprises to achieve structural transformation from the agricultural sector to the industrial sector

and from urban industry to the service sector, which will cause an increase in the efficiency of industry innovation in the national economy. This will further provide financial and technical support for enterprise production, focusing on relaxing private capital market access for productive service industries such as financial services, financial management, R&D innovation, and information networks to enhance urban innovation.

**Table 2.** Results of DID analysis.

| | **(1)** | **(2)** | **(3)** |
|---|---|---|---|
| **Variable** | *lntpatent* | *lntpatent* | *lntpatent* |
| *did* | 0.0770 ** | 0.2809 *** | 0.0631 * |
| | (2.03) | (5.49) | (1.70) |
| *lnpgdp* | | 0.2649 *** | 0.0817 |
| | | (4.40) | (0.84) |
| *lntrade* | | 0.1496 *** | 0.0691 ** |
| | | (9.36) | (2.35) |
| *edu* | | −0.0961 | 0.4189 *** |
| | | (−1.64) | (4.03) |
| *lnroad* | | −0.3779 *** | −0.0009 |
| | | (−10.56) | (−0.01) |
| *lngreen* | | −0.1373 *** | −0.0223 |
| | | (−5.39) | (−0.49) |
| *lndeposit* | | 1.1053 *** | 0.1608 |
| | | (37.93) | (1.08) |
| *_cons* | 4.7869 *** | −18.2840 *** | −2.8088 |
| | (487.03) | (−37.57) | (−1.19) |
| Time fixed effects | NO | NO | YES |
| Individual fixed effects | NO | NO | YES |
| N | 1789 | 1789 | 1789 |
| $R^2$ | 0.959 | 0.833 | 0.960 |
| F | 4.122 | 1581.102 | 6.847 |

Note: $t$ statistics in parentheses, * $p < 0.1$, ** $p < 0.05$, *** $p < 0.01$.

### 4.2. Parallel Trend Test

The DID method requires having no significant differences between the treatment and control groups prior to the pilot, or that the differences that do exist do not change sharply over time. For this reason, the parallel trend test needs to be re-run to ensure that the DID method used in this paper satisfies the parallel trend requirement. To this end, we refer to Nie et al.'s 2020 model-setting approach and add further dummy variables before and after the pilot policy to test for parallel trends [24]. Therefore, the following model was constructed for this purpose:

$$lntpatent_{i,t} = \beta_0 + \beta_1 before3_{i,t} + \beta_2 before3_{i,t} + \beta_3 before2_{i,t} + \beta_4 before1_{i,t}$$
$$+\beta_5 current_{i,t} + \beta_6 after1_{i,t} + \beta_7 after2_{i,t} + \beta_8 after3_{i,t} + \sum_{j=9}^{14} \alpha_j X_{i,t} + \mu_i + \varphi_t + \varepsilon_{i,t} \quad (4)$$

where *before4*, *before3*, *before2*, *before1*, *current*, *after1*, *after2*, *after3*, and *after4* are dummy variables that are observations for the 4 years before to 1 year before, the current year, and 1 year after to 4 years after whether the city entered the pilot policy or not, respectively. That is, the difference-in-differences term constructed in the previous section intersects with the pilot year dummy variables for the first 4 years to the first year, the current year, and the last year to the last 4 years. If most of the previously estimated coefficients are not significant or not very significant, this indicates that there is no significant difference between the treatment and control groups prior to the policy pilot, indicating that the parallel trend hypothesis is satisfied and the parallel trend test is passed.

The results in column (3) of Table 3 show that, after adding controls and controlling for both time and individual effects, none of the estimated coefficients for before are significant,

indicating that there is no systematic difference between the experimental and control groups before the pilot policy. The estimated coefficients for *after3* and *after4* start to be significant, indicating that a significant lifting effect is presented two years after the implementation of the new urbanization pilot policy. This indicates that the parallel trend hypothesis is satisfied, but there is also a lagged effect.

**Table 3.** Results of the parallel trend test and dynamic effects.

|  | (1) | (2) | (3) |
| --- | --- | --- | --- |
| **Variable** | *lntpatent* | *lntpatent* | *lntpatent* |
| *before4* | 0.5883 ** | 0.0792 | −0.0533 |
|  | (2.02) | (0.98) | (−0.84) |
| *before3* | 0.4375 | 0.0094 | −0.0099 |
|  | (1.54) | (0.09) | (−0.16) |
| *before2* | −0.0596 | −0.0250 | 0.0862 |
|  | (−0.30) | (−0.35) | (1.45) |
| *current* | 1.1422 *** | 0.3325 *** | 0.0594 |
|  | (4.15) | (3.60) | (1.13) |
| *after1* | 1.3471 *** | 0.4146 *** | 0.0898 |
|  | (4.90) | (4.20) | (1.54) |
| *after2* | 1.3038 *** | 0.2436 ** | 0.1008 |
|  | (4.41) | (2.32) | (1.54) |
| *after3* | 1.7896 *** | 0.3959 *** | 0.1088 * |
|  | (4.90) | (3.63) | (1.71) |
| *after4* | 1.7372 *** | 0.2343 ** | 0.2028 *** |
|  | (4.54) | (2.21) | (2.66) |
| Control variables | NO | YES | YES |
| Time fixed effects | YES | NO | YES |
| Individual fixed effects | YES | NO | YES |
| N | 1789 | 1789 | 1789 |
| $R^2$ | 0.065 | 0.834 | 0.960 |
| F | 12.797 | 794.174 | 4.290 |

Note: *t* statistics in parentheses, * $p < 0.1$, ** $p < 0.05$, *** $p < 0.01$.

### 4.3. Robustness Test

#### 4.3.1. PSM-DID

The selection of pilot cities for new urbanization is usually not random and is easily influenced by the city's economic development level, education level, and trade level, etc. Therefore, pilot cities may not be randomly assigned, thus causing selectivity bias. To reduce the sample selection bias, this paper uses the propensity score matching (PSM) method to select control group samples based on observable variables such as economic development level, education level, and trade level, and to reduce the self-selection bias of pilot cities for new urbanization construction. The PSM method first calculates the propensity score based on the factors affecting the selection of pilot cities for new urbanization construction, and then finds the control group sample with the closest propensity score for each sample in the treatment group, and uses a baseline regression model to estimate the average treatment effect between groups [16,18].

The matched treatment variables are dummy variables for whether the city is set as a pilot city for new urbanization, and the covariates are mainly year dummy variables (controlling for time effects), individual dummy variables (controlling for individual city effects), level of economic development (*lngdp*), level of trade (*lntrade*), level of education (*edu*), level of infrastructure development (*lnroad*), environmental quality (*lngreen*), and financial level (*lndeposit*). The matching methods used are kernel matching, radius matching, and nearest neighbor matching [25,26]. After matching is completed, observations that do not satisfy the common region assumption are removed.

The baseline regressions are performed again using the results estimated from the matched samples, and the estimation results in Table 4 are obtained, where (1) is the estimation result after kernel matching, (2) is the estimation result after radius matching, and (3) is the estimation result after nearest neighbor matching. It can be seen that the estimated coefficients of the difference-in-differences term *did* are all significantly positive at least at the 10% level after the inclusion of control variables, fixed time effects and individual effects, which is basically consistent with the previous regression results, indicating the robustness of the estimation results in this paper.

**Table 4.** Results of PSM-DID.

|  | **(1)** | **(2)** | **(3)** |
|---|---|---|---|
| **Variable** | *lntpatent* | *lntpatent* | *lntpatent* |
| *did* | 0.112 ** | 0.168 * | 0.129 ** |
|  | (2.44) | (1.97) | (2.29) |
| Control variables | YES | YES | YES |
| Time fixed effects | YES | YES | YES |
| Individual fixed effects | YES | YES | YES |
| N | 1789 | 443 | 1082 |
| $R^2$ | 0.967 | 0.968 | 0.969 |
| F | 4.204 | 2.518 | 3.158 |

Note: *t* statistics in parentheses, * $p < 0.1$, ** $p < 0.05$.

### 4.3.2. Counterfactual Test

The above test results indicate that the pilot policy of new urbanization construction has a significant impact on urban innovation, but it still needs to be further tested using the counterfactual method. In this paper, among 47 pilot cities, the year from 2011 to 2019 is randomly selected as the policy pilot time to construct the difference-in-differences term of the counterfactual. In this paper, 1000 samples are randomly selected, and the empirical *p*-value for each time is calculated and plotted as a scatter plot. If the empirical *p*-values obtained from 1000 random samples are mostly greater than 0.1 and the estimated results are significantly different from the estimated coefficients in Table 2, it means that it passes the counterfactual test and the estimation results of this paper are robust. As can be seen from Figure 1, the *p*-values obtained from the counterfactual test are mostly greater than 0.1 and differ significantly from the estimated value of *did* in Table 2, which is 0.0631, indicating that the estimation results of this paper are still robust.

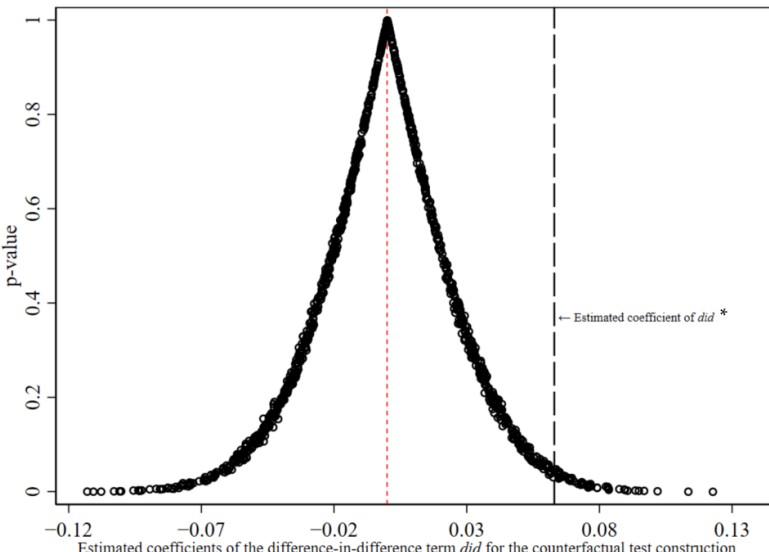

**Figure 1.** Estimated results of 1000 samplings using the Bootstrap method. * in Table 2.

## 5. Analysis of Heterogeneity and Influence Mechanism

### 5.1. Heterogeneity Analysis

The impact of pilot policies of new urbanization on urban innovation may be affected by urban heterogeneity due to the large differences in the characteristics of different cities. For this reason, further heterogeneity analysis is conducted in this paper.

(1)  Heterogeneity analysis of urban economic development levels.

This paper is based on a newly graded list of Chinese cities. This paper divides the research sample into first- and second-tier cities, third-tier cities, fourth-tier cities, and fifth-tier cities to analyze [27] the effects of new urbanization pilot policies on urban innovation under different city levels, and the estimation results are shown in Table 5. The results show that: Firstly, the pilot policy of new urbanization has the largest promotion effect on the urban innovation level of first- and second-tier cities, with a coefficient of 0.110 and significant at the 5% level. Secondly, the coefficients of both Tier 4 and Tier 5 cities are not significant. The reason for the above is that, on the one hand, it is the first- and second-tier cities themselves that have developed urban infrastructure, modern information technology, and human capital levels faster than the fourth- and fifth-tier cities, so they are more likely to have an effective innovation integration effect under the influence of the new urbanization pilot policy. On the other hand, Tier 4 and Tier 5 cities are underdeveloped cities, and the cities themselves have a lower level of technology, so pilot policy innovation is difficult for new urbanization.

**Table 5.** Estimated results of heterogeneity of cities at different levels.

| Variable | (1) First- and Second-Tier Cities | (2) Third-Tier Cities | (3) Fourth-Tier Cities | (4) Fifth-Tier |
|---|---|---|---|---|
| *did* | 0.110 ** | 0.090 * | 0.063 | −0.004 |
|  | (2.13) | (1.76) | (0.92) | (−0.02) |
| Control variables | YES | YES | YES | YES |
| Time fixed effects | YES | YES | YES | YES |
| Individual fixed effects | YES | YES | YES | YES |
| N | 153 | 567 | 891 | 178 |
| $R^2$ | 0.976 | 0.954 | 0.871 | 0.744 |
| F | 2.108 | 2.001 | 5.489 | 1.119 |

Note: *t* statistics in parentheses, * $p < 0.1$, ** $p < 0.05$.

(2)  Regional heterogeneity analysis.

Considering the large regional differences in China, cities with good economic development and a high level of science and education may be the first to pilot the policy, so regional heterogeneity is explored by eastern cities, middle cities, and western cities. The eastern, middle, and western regions are divided mainly according to the degree of economic development. Eastern cities mainly include cities in the following provinces: Beijing, Tianjin, Hebei, Liaoning, Shanghai, Jiangsu, Zhejiang, Fujian, Shandong, Guangdong, and Hainan. Middle cities mainly include cities in the following provinces: Shanxi, Inner Mongolia, Jilin, Heilongjiang, Anhui, Jiangxi, Henan, Hubei, and Hunan. Western cities mainly include cities in the following provinces: Sichuan, Chongqing, Guizhou, Yunnan, Tibet, Shaanxi, Gansu, Qinghai, Ningxia, Xinjiang, and Guangxi. It is important to note that the four Chinese municipalities have the same status as provinces. Hong Kong, Macau, and Taiwan are not included in the sample. The results of regional heterogeneity in Table 6 show that the policy effect of the new urbanization construction pilot policy is strongest in the western region, and the policy effect is not significant in both the eastern and central regions. The reason is that, during the implementation of the pilot policy, the western region has shown some latecomer advantages, and the urbanization rate has generally

increased faster in recent years, among which, in 2019, the western regions, such as Shaanxi, Guizhou, and Yunnan, have increased by more than 1%, which is faster than the national increase ratio in the same period.

**Table 6.** Regional heterogeneity results.

|  | (1) | (2) | (3) |
|---|---|---|---|
| Variable | Eastern Cities | Middle Cities | Western Cities |
| *did* | 0.056 | 0.026 | 0.229 ** |
|  | (1.09) | (0.48) | (2.02) |
| Control variables | YES | YES | YES |
| Time fixed effects | YES | YES | YES |
| Individual fixed effects | YES | YES | YES |
| N | 585 | 891 | 313 |
| R$^2$ | 0.975 | 0.953 | 0.921 |
| F | 6.932 | 6.712 | 2.295 |

Note: *t* statistics in parentheses, ** $p < 0.05$.

*5.2. Influence Mechanism Analysis*

The new urbanization construction may promote urban innovation development by enhancing human capital levels and injecting innovation into urban economic development. On the one hand, the new urbanization pilot policy construction includes providing training for employment for farmers and equal education for children of urbanized citizens to access compulsory education services; on the other hand, it provides free vocational training and entrepreneurship counseling for the unemployed. All these policies raise the level of human capital in the pilot cities; thus, the following model is constructed to test the mediation effect [28]:

$$hum_{i,t} = \varphi_0 + \varphi_1 did_{i,t} + \varphi_2 X_{i,t} + \mu_i + \lambda_t + \varepsilon_{i,t} \tag{5}$$

$$lntpatent_{i,t} = \gamma_0 + \gamma_1 did_{i,t} + \gamma_2 hum_{i,t} + \gamma_3 X_{i,t} + \mu_i + \lambda_t + \varepsilon_{i,t} \tag{6}$$

where *hum* is a measure of human capital and a mediating variable in this paper.

The results of the impact mechanism test are shown in Table 7. From the estimation results, the estimated coefficient of the double difference term in column (1) is significantly positive at the 1% level, which indicates that the pilot policy of new urbanization construction significantly enhances the human capital level of cities. Meanwhile, the estimated coefficient of the difference-in-differences term *did* in column (2) is still significant and the estimated coefficient of *hum* is still significantly positive at the 10% level, indicating a strong mediating effect, which indicates that human capital is a mediating variable of the new urbanization construction pilot policy to promote urban innovation.

**Table 7.** Results of the influence mechanism.

|  | (1) | (2) |
|---|---|---|
| **Variable** | **hum** | **lntpatent** |
| *did* | 0.476 *** | 0.059 ** |
|  | (4.33) | (2.05) |
| *hum* |  | 0.009 * |
|  |  | (1.81) |
| Control variables | YES | YES |
| Time fixed effects | YES | YES |
| Individual fixed effects | YES | YES |

**Table 7.** *Cont.*

| Variable | (1) hum | (2) lntpatent |
|---|---|---|
| N | 1791 | 1789 |
| R$^2$ | 0.967 | 0.955 |
| F | 35.111 | 6.097 |

Note: *t* statistics in parentheses, * $p < 0.1$, ** $p < 0.05$, *** $p < 0.01$.

## 6. Conclusions and Policy Implications

Based on the panel data of 199 cities in China from 2011 to 2019, this paper empirically analyzes the impact of pilot policies of new urbanization construction on urban innovation using a double difference method. The study shows that: (1) New urbanization pilot policies can effectively improve urban innovation. (2) Heterogeneity analysis shows that there is significant city-level heterogeneity and regional heterogeneity in the improvement of urban innovation levels through new urbanization pilot policies. The impact effect of new urbanization pilot policies in first- and second-tier cities is higher than that in fourth- and fifth-tier cities; the effect of new urbanization pilot policies in western regions is higher than that in eastern and middle regions. (3) The analysis of the impact mechanism indicates that the new urbanization pilot policies enhance urban innovation through human capital.

Based on the above research findings, this paper puts forward the following policy recommendations:

(1) New urbanization has an enhancing effect on urban innovation. Therefore, the Chinese government should promote the construction of new urbanization, reshape the concept of urbanization development, and improve the level of human capital and infrastructure. The successful experience of the pilot policies of new urbanization should be summarized and widely promoted so that more regions can enjoy the economic development effects brought by new urbanization.

(2) Different cities should develop differently in the process of new urbanization. On the one hand, cities and regions with low comprehensive strength should continue to explore their own development potential, vigorously promote the construction of new urbanization, and increase the construction of related industries and scientific research infrastructures. They should also explore digital financial technology investment and radiate the economic and digital financial technology benefits brought on by the construction of new urbanization to all aspects of urban residents' lives and enterprises' production, so as to promote the construction of local urbanization to a high-quality development stage. On the other hand, cities and regions with well-developed comprehensive economic strength should, on the basis of maintaining their original advantages, fully exploit other economic advantages brought on by new urbanization, such as actively developing high-tech industries, cultivating new economic growth points, and forming unique industrial advantages of cities.

(3) New urbanization affects urban innovation through human capital. For this reason, city managers should focus on cultivating human capital development in the process of new urbanization construction, seize the opportunity of new urbanization policies to promote urban economic transformation and upgrading, and provide human capital levels.

**Author Contributions:** Study design, Y.W. and H.D.; data processing, Y.W., S.Y., S.L. and D.Y.; statistical analysis, Y.W., H.D., S.Y., S.L. and D.Y.; writing—review and editing, Y.W., H.D., S.Y., S.L. and D.Y. All authors have read and agreed to the published version of the manuscript.

**Funding:** This research was funded by the National Natural Science Foundation of China, grant number 71761015.

**Institutional Review Board Statement:** Not applicable.

**Informed Consent Statement:** Not applicable.

**Data Availability Statement:** Data for this work are available through the corresponding author.

**Acknowledgments:** The authors are grateful to the anonymous reviewers for their valuable comments, and the authors are grateful to the editors for their hard work.

**Conflicts of Interest:** The authors declare no conflict of interest.

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
