# Peer review of "The Effects of New Urbanization Pilot City Policies on Urban Innovation: Evidence from China"

_sustainability, doi:10.3390/su151411352_

Round 1

Reviewer 1 Report

Dear authors,

Thank you very much for this paper, which I enjoyed reading. Although I like your research idea and I also acknowledge the relevance of the research problem, I see some shortcomings in the paper, which I would like to point out in the following.

First of all, I notice that the paper is only interested in deriving practical recommendations for action and neglects the scientific part, especially in terms of theory application and further development. The theoretical relationship between patent applications and the pilot policy of new urbanization construction has not become clear to me. For clarification, you should theoretically derive the hypothesis for the study and discuss it against the background of the relevant literature (which you already do to some extent in the introduction).

In the methodology section, you should also better explain the selection of variables for estimation. What theoretical motivations led you to include certain variables as controls? Is there relevant research literature you can draw on? In short, I would recommend hypothesis generation based on a literature review.

I would also have expected more background information in the presentation of the data, such as a correlation table. I assume that some variables may be highly correlated. Also, I have a feeling that the direction of causality is not clear for some variables. In regional innovation research, these problems are discussed very often. How did you address this issue in your research?

With regard to your empirical analysis, I also note that you perform many test procedures to prove the validity of your results. However, from my point of view, the first question to ask is whether you can show a statistically relevant result at all. The variable of interest (did) is only significant at the 10 percent level in your panel estimate when time and individual fixed effects are included (see Table 2). Thus, the significance level is very low. I missed a critical statement on this in your article. Furthermore, the results should also be discussed theoretically and placed in the context of the research literature. A coherent theoretical framework would be helpful for this objective and would also benefit the whole paper.

Overall, I would also have expected the novelty and originality of your findings to be better presented. What is the added value of your empirical results for research?

I hope that my comments can help to further improve the quality of the work.

Author Response

Comment 1: I notice that the paper is only interested in deriving practical recommendations for action and neglects the scientific part, especially in terms of theory application and further development. The theoretical relationship between patent applications and the pilot policy of new urbanization construction has not become clear to me. For clarification, you should theoretically derive the hypothesis for the study and discuss it against the background of the relevant literature (which you already do to some extent in the introduction).

Response: Many thanks to the reviewer's comments, we have added a discussion of the results of new urbanization policies affecting urban innovation in section 4.1, which further analyzes why new urbanization policies affect urban innovation. The specific modifications are as follows:

Why new urbanization construction pilot policies have increased the level of urban innovation? The promotion effect of new urbanization on urban innovation capacity mainly comes from three aspects: first, the construction of industrial science and technology innovation parks etc. in the new urbanization construction positively influences the production and operation behavior and infrastructure support of micro enterprises, providing the basic conditions for innovation of enterprises in the region. Second, the focus of new urbanization construction also includes developing economies of scale and reducing transaction costs as well as trade costs, such as new urbanization promoting the agglomeration and diffusion of factors through the role of population mobility and resource allocation (Shen et al., 2022), and improving urban innovation capacity through improving labor quality. Third, new urbanization guides the gradual development of industries in the region from traditional labor-intensive to modern capital-led (Li et al., 2020), i.e., it guides enterprises to achieve structural transformation from the agricultural sector to the industrial sector and from urban industry to the service sector, which will cause an increase in the efficiency of industry innovation in the national economy. This will further provide financial and technical support for enterprise production, focusing on relaxing private capital market access for productive service industries such as financial services, financial management, R&D innovation, and information networks to enhance urban innovation.

Comment 2: In the methodology section, you should also better explain the selection of variables for estimation. What theoretical motivations led you to include certain variables as controls? Is there relevant research literature you can draw on? In short, I would recommend hypothesis generation based on a literature review.

Response:

Thank you for your valuable comments. We have added a description of the variables.

Based on the approach of the available literature, the control variables affecting urban innovation were selected as follows.

Level of urban economic development (lngdp): measured using the natural logarithm of the city's GDP (Zhou & Li, 2022). Since the level of economic development represents the economic strength of a region, if the level of economic development is higher, it is more likely to increase regional infrastructure investment and R&D investment, which will have a positive promotion effect on the level of urban innovation.

Trade level (lntrade): measured using the natural logarithm of the total domestic and foreign trade of each city. The development of international trade has greatly contributed to scientific and technological innovation (Spulber, 2008), which has been one of the most important factors for China to gain a foothold in international cooperation since its accession to the World Trade Organization.

Education level (edu): measured using years of schooling per capita in each city. Education can provide talents and intellectual resources for science and technology innovation. China's higher education and research institutions have produced a large number of scientific and technological talents who have become an important force for scientific and technological innovation. In addition, education can provide a wealth of disciplinary knowledge and skills, which in turn can provide the host city with innovation capital and promote urban innovation.

Infrastructure development (lnroad): measured using the natural log of road miles owned by each city. The public services carried by infrastructure are at the heart of cities. Taking the transportation construction in Beijing, Shanghai and the Guangdong-Hong Kong-Macao Greater Bay Area as examples, infrastructure innovation has become an important force in China's national innovation system(Lu et al., 2022).

Environmental quality (lngreen): measured using the natural logarithm of green space per city. The Chinese government has proposed a policy to protect the environment, and in the process of protecting the environment, environmental technology has integrated green technologies into various industries and fields, and the application of information technology, cloud computing and big data in the field of environmental protection continues to expand and deepen, promoting technological innovation in cities. In addition, the continuous integration of environmental technology and new technologies further drives the great development of the environmental protection industry.

Financial level (lndeposit): measured using the natural logarithm of resident savings in each city. The development of technology enterprises cannot be separated from the support of financial capital, which provides capital support for urban innovation.

Comment 3: I would also have expected more background information in the presentation of the data, such as a correlation table. I assume that some variables may be highly correlated. Also, I have a feeling that the direction of causality is not clear for some variables. In regional innovation research, these problems are discussed very often. How did you address this issue in your research?

Response: Thank you for your suggestions. We have added more specific explanations for the selection of variables and supplemented the correlation coefficient tables and multicollinearity tables as validation of the relationships between the variables. Also, a discussion of the results has been added in section 4.1.

Comment4 : With regard to your empirical analysis, I also note that you perform many test procedures to prove the validity of your results. However, from my point of view, the first question to ask is whether you can show a statistically relevant result at all. The variable of interest (did) is only significant at the 10 percent level in your panel estimate when time and individual fixed effects are included (see Table 2). Thus, the significance level is very low. I missed a critical statement on this in your article. Furthermore, the results should also be discussed theoretically and placed in the context of the research literature. A coherent theoretical framework would be helpful for this objective and would also benefit the whole paper.

Response: Thank you for your suggestion. Although the significance of did in this paper is 10%, it still indicates that there is a facilitating effect, and also considering the new type of urbanization as a long-term macro policy in China, this policy is not fully effective yet due to the data time span problem, and also provides policy space for future China to continue to expand the new type of urbanization city pilot with practical guidance. We also compared the CNKI literature on the impact of China's new urbanization policies on other aspects and found that there was also a 10% significance. Also results discussion been added in section 4.1.

Comment5 : I would also have expected the novelty and originality of your findings to be better presented. What is the added value of your empirical results for research?

Response: Thank you for your valuable comments. Based on the existing writing, we have added a description of the deserving value of new urbanization. The additions are as follows:

The value of this paper lies in the following three points: (1) This study can enrich our understanding of new urbanization, accurately grasp the synergistic requirements of China's new urbanization and urban innovation, and provide important policy references for the enhancement of China's urban innovation capacity. (2) On the basis of theoretical and empirical studies, it provides policy recommendations for further promoting the construction of new urbanization and focusing on its economic effects. In the context of the new era, how to enhance the innovation capacity of cities and promote high-quality economic development has become an important practical issue. By promoting the construction of new urbanization, stimulating the inherent potential of new urbanization and improving infrastructure construction, etc., the Chinese government will be an important way to improve the innovation capacity of cities. (3) This paper can formulate the key development directions of new urbanization policies for the improvement of urban innovation capacity in different regions, and further provide realistic and feasible guidance for the improvement of overall innovation capacity.

Reviewer 2 Report

1. The first paragraph of the introduction should be rewritten, the data should be updated, and the references should cite the latest research results.

2. The structure of the “Introduction” should be reorganized and rearranged with relevant content.

3. The literature review section is lacking, and a comprehensive review of the previous literature is needed to know the significance of the existence of the article and its innovations.

4. Lack of discussion of the results of the findings, no DISCUSSION .

5. Lack of background explanation on some pilot cities and policies.

6. The timeliness is poor, it's already 2023 and the data is only up to 2019, is it because of the epidemic, it's better to have a description.

7. The references in the article do not cite the latest research results, are not very innovative, and have an incomplete structure.

General language writing

Author Response

Comment1 : The first paragraph of the introduction should be rewritten, the data should be updated, and the references should cite the latest research results.

Response:Thanks to the suggestion, we have rearranged the first paragraph and added new references.

Comment2 :The structure of the “Introduction” should be reorganized and rearranged with relevant content.

Thank you for your comments. We have divided the introduction section into three subsections: introduction, purpose of the study, and significance of the study, which makes the structure of the introduction clearer.

Comment3 :The literature review section is lacking, and a comprehensive review of the previous literature is needed to know the significance of the existence of the article and its innovations.

Response: Thank you for your constructive suggestions. Because the topic of the study is the Chinese issue, we have added the following in comparison with the Chinese literature. New urbanization affects innovation has become a preliminary consensus in academic circles, but the degree of impact and mechanism of new urbanization innovation is not clear. The mechanism through which new urbanization affects the increase of innovation remains a hot topic of debate. The question of whether a series of policy measures adopted during the promotion of China's new urbanization city pilot policy have effectively increased urban innovation is subject to further study. The following shortcomings remain in existing studies:

First, most of the existing domestic studies use national and provincial-level regional data to measure new urbanization and impact factors, which can hardly reflect the full picture of new urbanization in China and ignore the impact of new urbanization on micro-cities, resulting in a lack of detailed analysis of impacting subjects and impact mechanisms.

Second, many literatures ignore the policy practice effects brought by pilot cities of new urbanization, which serve as a realistic practice channel to enhance urban innovation and provide quasi-natural experiments to further verify the realistic impacts of new urbanization, thus providing corresponding policy suggestions to continue deepening the construction of new urbanization, optimizing resource allocation and improving economic efficiency.

Comment4: Lack of discussion of the results of the findings, no DISCUSSION .

Response: Thank you for the valuable comment. A discussion of the results has been added in section 4.1.

Comment5 : Lack of background explanation on some pilot cities and policies.

Response: Thanks to your comments, we have added more detailed background on the pilot cities and policies. Specific additions include:

After the reform and opening up in 1978, China ushered in a wave of industrialization centered on economic construction, which led to the rapid development of industry and commerce and greatly accelerated the process of urbanization. At the same time, China's urbanization shifted from encouraging the development of small and medium-sized cities and small towns to full-scale urban expansion. Existing studies have made a lot of detailed research on the content and characteristics of traditional urbanization, and it is found that the period from reform and opening up to 2012 is called the traditional urbanization stage. The prominent feature of traditional urbanization is the rapid increase of urbanization rate, which has reached 52.57% during the traditional urbanization stage, nearly tripled by 34.67%, and China's urbanization has achieved rapid development and made significant contributions to the historical process of promoting urbanization in the world. Along with the rapid growth of urbanization rate, it has also produced a series of negative impacts on economy and society. For example, the one-sided focus on urban expansion, blindly engaging in "city building" and building new districts has led to low quality urbanization, urbanization level lower than the level of economic and industrial development, urbanization of land faster than urbanization of population, and structural problems such as high pressure on transportation network and poor local traffic congestion.

If the traditional urbanization development model continues to be adopted, it will further aggravate and trigger economic and social hidden dangers such as environmental degradation, weak consumer demand, and sloppy and inefficient types of investment in China. Therefore, there is an urgent need to draw on advanced international experience, accelerate the high-quality development of urbanization, take a new type of people-oriented urbanization path, and fully optimize and upgrade traditional urbanization. To this end, since 2015, China has set up three batches of new-type urbanization comprehensive reform pilots to explore the transformation path of urbanization from focusing on quantitative scale increase to quality connotation improvement. The Chinese government has successively introduced relevant policies to support and accelerate the construction of new urbanization over the past six years. As of 2020, a total of 188 cities (districts, counties and towns) have been promoted to implement new urbanization in three batches of pilot cities, and the policy of comprehensive pilot cities for new urbanization is beginning to bear fruit, highlighted by the fact that new urbanization has abandoned the past pursuit of simple urban population ratio increase and scale expansion, and has achieved innovation and innovation in various aspects such as concepts, institutions and culture.

Comment6 :The timeliness is poor, it's already 2023 and the data is only up to 2019, is it because of the epidemic, it's better to have a description.

Response: Thank you for your advice. The main reasons for selecting data 2011-2019 are: first, the main study of this paper is to do experimental evaluation of new urbanization as a policy, and the data are selected to cover the years of new urbanization policy, which is consistent with the specification of the experimental DID method. Second, because the data study units in this paper are Chinese prefecture-level municipalities and above, the data years are selected to 2011-2019, considering the accessibility of control variables and explanatory variables to ensure that all variables are available. Third, due to the impact of the epidemic, China's epidemic prevention policy adopted a national quarantine policy, and the new urbanization construction was also affected by a certain stagnation at this time, and the period of the epidemic was not included in the study considering the special period of the epidemic.

Comment7 : The references in the article do not cite the latest research results, are not very innovative, and have an in recomplete structure.

Response: We have added the latest references.

Reviewer 3 Report

I read the manuscript entitled “The Effects of New Urbanization Pilot City Policies on Urban Innovation: Evidence from China” thoroughly. I think that the subject of the article is interesting. However, I would like to suggest some comments as follows:

1. The conclusion in the abstract is too narrow and localized, and should emphasize more generalized deductions.

2. In the introduction, the authors mainly concentrated on local (Chinese) resources, while it is expected to be of more international interest. I suggest that the introduction enriched by adding some novel and critical literature to the review part of the introduction.

3. The method and results are well written and justified.

4. Some points are considerable in the discussion, as follows: First, the authors should try to provide more generalized outcomes of the research at the end of the discussion. Second, the authors indicate to the term of Urban Innovation, but what is their definition of Urban Innovation remained neglected. Third, the outcomes of the research should be compared with similar works. Fourth, are “Urbanization Pilot City Policies” faced with challenges or shortages? This should be indicated in the discussion.

5. I suggest the authors provide a more concise conclusion by pointing out the generalized outcomes of the research and indicating the implications.

6. I have no more comments.

Author Response

Thank you very much for your approval of our paper.

Comment1 :The conclusion in the abstract is too narrow and localized, and should emphasize more generalized deductions. In the introduction, the authors mainly concentrated on local (Chinese) resources, while it is expected to be of more international interest. I suggest that the introduction enriched by adding some novel and critical literature to the review part of the introduction. I suggest the authors provide a more concise conclusion by pointing out the generalized outcomes of the research and indicating the implications.

Response: Thank you for your advice. Add the following on the current status of international urbanization development and the general applicability of the conclusions:

In the literature, one hand, the evolution of urbanization cannot be separated from industrialization, such as the first industrial revolution and the second industrial revolution drove the urbanization process in Britain and the United States, respectively (Feinstein, 1998). On the other hand, urbanization also drives economic development and becomes an important spatial vehicle for regional economic activities, such as the London metropolitan area, the Tokyo metropolitan area, the Bosworth urban agglomeration, and the Beijing metropolitan area in China. It has been proved that urbanization is a dynamic evolutionary process embedded in the national economic system, and the pattern and level of urbanization are determined by the characteristics of economic development stages and institutional policy system (Davis & Henderson, 2003). The urbanization in Britain was based on the development of the industrial revolution and the lessons learned from the enclosure movement before the industrial revolution, and regulations were enacted to steadily promote urbanization. On the contrary, some Latin American countries have experienced "over-urbanization" in which urbanization has greatly exceeded the level of industrialization and agricultural modernization. In Mexico, Chile, Argentina, Uruguay and other countries, the urbanization rate has reached 80% or even 90%, but this is a phenomenon of "false urbanization".

Therefore, this paper proposes a direction for urbanization development in less developed countries or regions, which is to follow the natural laws of urban evolution, improve public infrastructure and public services in cities, and avoid "over-urbanization" and "lagging urbanization". The promotion of new urbanization should be matched with the current economic development, taking the level of economic development as the basis for urbanization, focusing on assessing whether public services are coordinated with the needs of urban population, and avoiding the speed of upgrading infrastructure and public services lagging behind the speed of urbanization.

Comment2 :Some points are considerable in the discussion, as follows: First, the authors should try to provide more generalized outcomes of the research at the end of the discussion. Second, the authors indicate to the term of Urban Innovation, but what is their definition of Urban Innovation remained neglected. Third, the outcomes of the research should be compared with similar works. Fourth, are “Urbanization Pilot City Policies” faced with challenges or shortages? This should be indicated in the discussion.

Response: Thanks to your suggestion, we have added an explanation of urban innovation so that readers can better understand the meaning of urban innovation in this paper.

Urban innovation capability is a manifestation of innovative cities (Wang & Deng, 2022), which refers to cities with strong independent innovation capability, outstanding role in supporting and leading science and technology, high level of sustainable economic and social development, and significant regional radiation and driving effect. As a regional innovation highland, innovative cities play a crucial role and are important pillars for the construction of innovative countries. At present, innovative cities in China are mainly oriented to prefecture-level cities and are often judged by an evaluation index system consisting of five primary indicators and 30 secondary indicators, such as innovation governance, original innovation, technology innovation, achievement transformation and innovation driving force. In this paper, the total number of patent applications is selected as the measurement index, which can reflect the innovation power of the city.

Round 2

Reviewer 1 Report

Dear Authors,

Thank you very much for the various changes in your manuscript, which I appreciate very much. I think the work as a whole has benefited greatly. My initial concerns have been addressed by this, so I no longer have any objections to publication.

Thank you again for the well done revision.

Author Response

Thank you for your approval of our paper and your valuable comments made it possible to meet the publication requirements.

Reviewer 2 Report

The article is well revised, but the format still needs to be modified to meet the requirements of the journal. The literature review could be a little more complete, and some similar research recommendations could be appropriately supplemented.  Some ideas in the literature can be referred to, such as:  DOI: 10.3390/su14105765; 

Policy Implications: The policy part should be closely linked to the conclusion of the article, but also need to think about it seriously.

Overall, it was OK.

Author Response

Thank you for your constructive suggestions, and we have revised the format to meet the journal's requirements. We referred to the paper you recommended and ended up revising the policy implications as well.